# Cell-specific synaptic plasticity induced by network oscillations

**Shota Zarnadze[1†], Peter Bäuerle[1†], Julio Santos-Torres[1], Claudia Böhm[2], Dietmar Schmitz[2,3,4], Jörg RP Geiger[1,3], Tamar Dugladze[1,3‡], Tengis Gloveli[1,4*‡]**

[1]Institute of Neurophysiology, Charité -Universitätsmedizin Berlin, Berlin, Germany; [2]Neuroscience Research Center, Charité -Universitätsmedizin Berlin, Berlin, Germany; [3]The NeuroCure Cluster of Excellence, Berlin, Germany; [4]Bernstein Center for Computational Neuroscience, Berlin, Germany

**Abstract** Gamma rhythms are known to contribute to the process of memory encoding. However, little is known about the underlying mechanisms at the molecular, cellular and network levels. Using local field potential recording in awake behaving mice and concomitant field potential and whole-cell recordings in slice preparations we found that gamma rhythms lead to activity-dependent modification of hippocampal networks, including alterations in sharp wave-ripple complexes. Network plasticity, expressed as long-lasting increases in sharp wave-associated synaptic currents, exhibits enhanced excitatory synaptic strength in pyramidal cells that is induced postsynaptically and depends on metabotropic glutamate receptor-5 activation. In sharp contrast, alteration of inhibitory synaptic strength is independent of postsynaptic activation and less pronounced. Further, we found a cell type-specific, directionally biased synaptic plasticity of two major types of GABAergic cells, parvalbumin- and cholecystokinin-expressing interneurons. Thus, we propose that gamma frequency oscillations represent a network state that introduces long-lasting synaptic plasticity in a cell-specific manner.

**\*For correspondence:** tengis.gloveli@charite.de

[†]These authors contributed equally to this work
[‡]These authors also contributed equally to this work

**Competing interests:** The authors declare that no competing interests exist.

## Introduction

Neural oscillations are thought to play an important role in learning and memory processing (*Axmacher et al., 2006*; *Düzel et al., 2010*; *Nyhus and Curran, 2010*). Learning is based on activity-dependent modification of synaptic strength in order to incorporate transient experiences into persistent memory traces (*Citri and Malenka, 2008*). Gamma-band oscillations and sharp wave-ripple activity (SWR), involved in memory encoding (*Jutras and Buffalo, 2010*) and consolidation (*Buzsáki, 1989*; *Girardeau et al., 2009*; *Jadhav et al., 2012*), respectively, appear to be interlinked in the course of memory processing. However, to date, it had been unclear whether gamma rhythms itself represent a network state that can directly promote the formation of long-lasting synaptic plasticity within the cortical network.

The hippocampus is an important structure for memory acquisition, consolidation and spatial orientation (*Buzsáki and Moser, 2013*; *Eichenbaum and Cohen, 2014*). Within the hippocampus, the CA3 area constitutes an autoassociative neural network, in which three pathways converge: mossy fibers and associational-commissural (A/C) and perforant path (PP) projections. Each of these pathways display plasticity (*Berzhanskaya et al., 1998*; *Malenka and Bear, 2004*; *Nicoll and Schmitz, 2005*), which has seen the CA3 area harnessed as a well-suited and popular model for studying activity-dependent modification of synaptic transmission. However, hippocampal neural plasticity phenomena have traditionally been studied using tetanus-, pairing- or chemically induced plasticity-protocols, while the applicability of network oscillations, such as used in our study, as investigative tool had been unproven to date.

**eLife digest** Changes in the strength of synapses – the connections between neurons – form the basis of learning and memory. This process, which is known as synaptic plasticity, incorporates transient experiences into persistent memory traces. However, a single synapse should not be viewed in isolation. Neurons typically belong to extensive networks made up of large numbers of cells, which show coordinated patterns of activity. The synchronized firing of the neurons in such a network is referred to as a network oscillation.

The frequency of an oscillation – that is, the number of times per second that its component cells are active at the same time – reflects distinct physiological functions. For example, high frequency oscillations called gamma waves help new memories to form, but it is not clear exactly how they do this. By studying gamma oscillations in a brain region called the hippocampus, Zarnadze, Bäuerle et al. provide insights into the underlying mechanisms.

Signals from "excitatory" neurons make the neuron on the other side of the synapse more likely to fire in response, and signals for "inhibitory" neurons make it less likely to fire. By recording the activity of excitatory neurons in mouse brain slices, Zarnadze, Bäuerle et al. show that gamma oscillations increase the strength of excitatory synapses in the hippocampus, allowing neurons to signal more easily across these connections. Blocking the activity of a protein called metabotropic glutamate receptor 5 prevents this increase in excitatory synaptic strength, suggesting that these receptors play an important role in memory processing. In contrast to excitatory neurons, gamma oscillations have different effects on two types of inhibitory neurons within the hippocampus. The oscillations increase the excitability of gamma-supporting inhibitory neurons, but at the same time reduce that of gamma-disturbing inhibitory neurons. These opposing changes in turn support synaptic plasticity.

By showing that gamma oscillations contribute to changes in synaptic strength within the hippocampus, Zarnadze, Bäuerle et al. help to explain the importance of these rhythms for memory processing. Further research is now needed to fully decipher the roles of different cell types, and the synaptic connections between them, in the formation of new memories.

GABAergic interneurons (INs) have been shown to play a major role in controlling oscillatory activity, as well as synaptic transmission and plasticity in cortical networks (*Ben-Ari, 2006*; *Klausberger and Somogyi, 2008*). Major inhibitory cell types, parvalbumin- (PV) and cholecystokinin (CCK)-expressing INs provide distinct forms of inhibition and have complementary roles in cortical circuits. In fact, PV-expressing INs are assumed to act as "fast signaling devices" (*Jonas et al., 2004*) and provide precisely-timed phase-modulated inhibition to control the timing of discharge in individual neurons as well as the synchronization and emergence of oscillations at the level of the network (*Cobb et al., 1995*; *Gloveli et al., 2005a*; *Bartos et al., 2007*; *Sohal et al., 2009*). In contrast, CCK-expressing INs show a slower but variable discharge pattern and asynchronous GABA release (*Hefft and Jonas, 2005*; *Daw et al., 2009*). High expression of various receptors for neuromodulators (*Freund, 2003*; *Armstrong and Soltesz, 2012*) post- and presynaptically, suggests that these INs may regulate excitability in the network by mediating inhibition in a behavioral state-dependent manner.

We investigated the interaction and interdependence of two classical hippocampal network patterns, gamma frequency oscillations and SWRs, important in memory encoding and consolidation, respectively. We found that in vivo theta-nested gamma oscillations have an enhancing effect on subsequent SWRs in awake behaving mice. Analysis of the underlying molecular, cellular and synaptic mechanisms in vitro slice preparations showed changes in SWR-associated excitatory synaptic strength between pyramidal cells (PC) that are mediated postsynaptically and depend on metabotropic glutamate receptor-5 (mGluR5) activation. In stark contrast to excitation, alteration of inhibitory synaptic strength was independent of postsynaptic activation and less pronounced, reflecting an IN-specific, directionally biased synaptic plasticity, as demonstrated in our study for two major GABAergic inhibitory cell types, PV- and CCK-expressing INs. Our results suggest that gamma frequency oscillations represent a network state that promotes the formation of long-lasting synaptic plasticity in the hippocampal area CA3, leading to modification of synaptic strengths in a cell-specific manner.

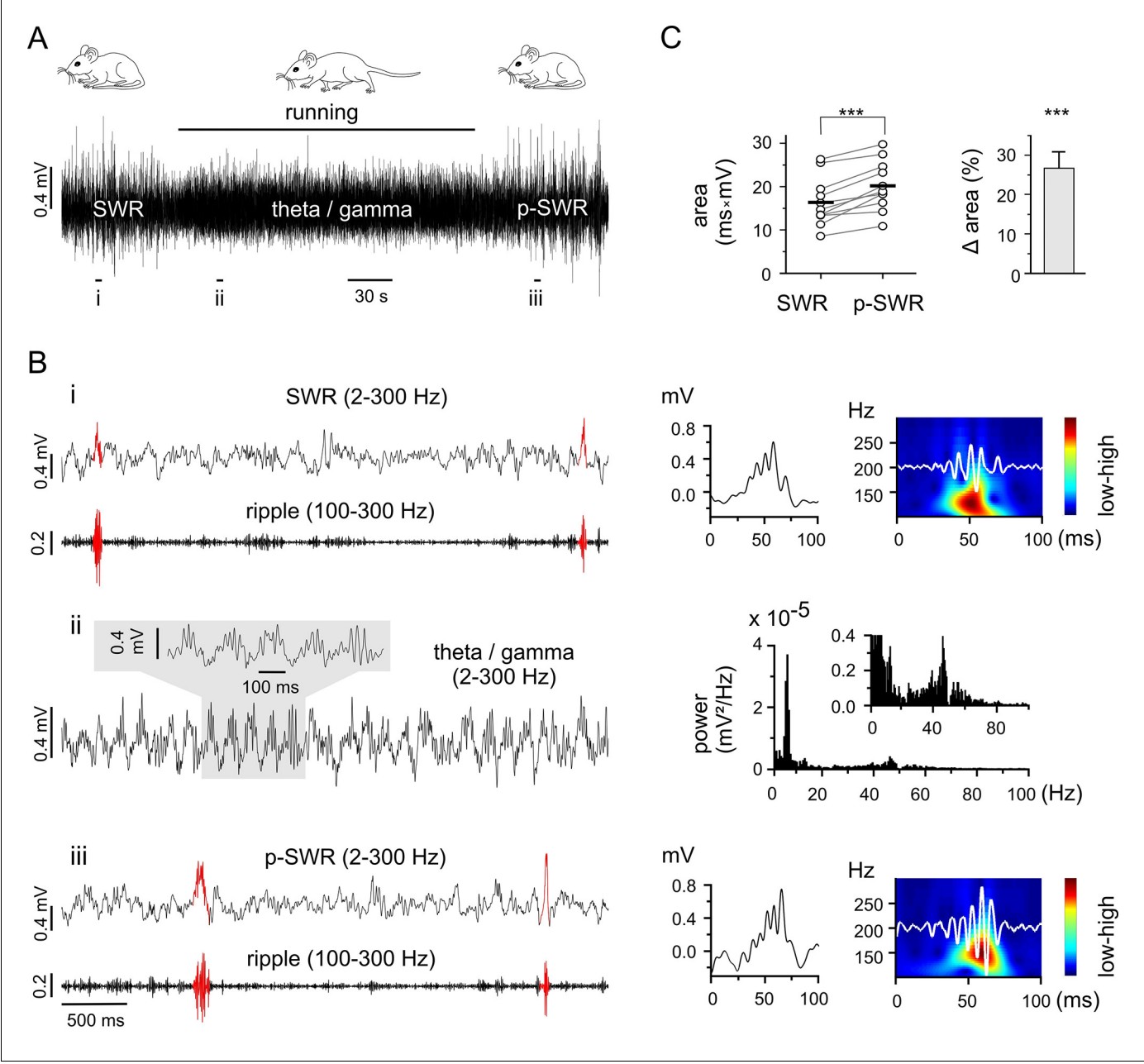

**Figure 1.** Theta-nested gamma rhythms enhance hippocampal SWRs in vivo. (**A**) Representative LFP recording in an awake mouse illustrates the occurrence of different network states in a behavioral-dependent manner. The initial spontaneous SWRs during a quiet state (left, SWR) are replaced by a running-associated theta-nested gamma rhythm (middle), followed by another rapid reversal to SWRs (right, p-SWR). Note the higher amplitude of the p-SWRs. Prolonged running period is marked by the black bar above. (**B**) (i, ii, iii) Left, three example excerpts from the trace in (**A**) at higher temporal resolution. (i) Left, the initial quiet state example band-pass filtered at 2–300 Hz and 100–300 Hz to illustrate the SWRs and the corresponding ripple component, respectively. Two SWRs are accentuated in red. Right, a single SWR together with its wavelet transform (color-coded power spectral density with superimposed corresponding ripple trace in white). (ii) Left, the theta-nested gamma example with a small excerpt shown above. Right, the corresponding spectral analysis demonstrates the predominant theta (7.3 Hz) and gamma (46.3 Hz) peak. (iii), the p-SWR with the same type of illustration as for the SWR shown in (i). (**C**) Left, a direct comparison of the mean values of SWR areas before and after the intervening gamma episode (n = 11, 4 mice) highlight a significant increase in SWR areas (p=0.0006, Wilcoxon signed rank test). The respective grand means are indicated by the horizontal bold bars. Right, the corresponding percentage increase of mean SWR areas.

## Results

### Theta-nested gamma frequency oscillations reinforce subsequent SWRs in awake behaving mice

We investigated two major context-dependent activity patterns, SWRs and gamma frequency oscillations, in awake behaving mice. Using local field potential (LFP) recordings from dorsal hippocampus we found spontaneous SWRs in resting states of quietly sitting mice, while running behavior was accompanied by theta-nested gamma oscillations (*Figure 1A*). As both rhythms have been proposed as closely linked to memory processing (*Axmacher et al., 2006*; *Girardeau et al., 2009*; *Jutras and Buffalo, 2010*; *Jadhav et al., 2012*), we studied the general interdependence of these two network patterns in a behavioral paradigm.

As the two rhythms are each correlated with separate behavioral states (resting or running), control of behavioral expression is a means of targeting the corresponding network pattern (*Figure 1A and B*). Thus, we used prolonged running episodes, at an average of 3 min, (mean 3:06 ± 0:32 min, n = 11, 4 mice; range from 2:28 to 4:30 min depending on the actual running performance), to investigate the interaction between the two network patterns. The impact of running-associated theta-nested gamma frequency oscillations on subsequent SWRs was evaluated by comparing the areas of SWRs in the two-minute time windows directly preceding and following a theta-nested gamma episode. The post-gamma SWR (p-SWR) areas exhibited significantly enlargement at an average of 25.7% (n = 11, p=0.0006, *Figure 1C*) and no significant change in frequency (SWR: 0.18 ± 0.10 Hz; p-SWR: 0.16 ± 0.10 Hz, n = 11, p=0.23). This result indicates an enhanced network activity and suggests a surprisingly direct effect of the running associated theta-nested gamma oscillations. But the fact that the internal state might not be fixed for an extended time period hampered conclusive interpretation of this result. In particular, the altered p-SWR could also reflect a change of the animals internal state including the contribution of a different set of cell assembles (*Buzsáki, 2015*), which is difficult to control in vivo. Consequently, in order to carry out the experiments under well-controlled conditions with the option to investigate the underlying mechanism in detail, we switched to a well-established in vitro model.

### Gamma frequency oscillations promote long-lasting network changes in acute slice preparations

We subsequently investigated the synaptic, cellular and molecular mechanisms of gamma oscillation-induced effects on SWR activity in in vitro acute hippocampal slices, a model that permits the reproduction of both oscillatory network patterns (*Gloveli et al., 2005a*, *2005b*; *Maier et al., 2009*; *Dugladze et al., 2012*).

Prompted by our in vivo results, we investigated the interaction and interdependence of gamma frequency oscillations and SWRs by monitoring LFPs in the stratum pyramidale of the hippocampal area CA3. In good agreement with our in vivo results, SWRs and gamma frequency oscillation patterns represented two 'competing', mutually exclusive network states in vitro: spontaneously occurring SWRs (mean frequency: 1.33 ± 0.10 Hz, n = 30) disappeared shortly (31.0 ± 2.8 s) after bath application of kainic acid (KA, 400 nM) and reappeared within a few minutes (14.6 ± 0.7 min) after KA washout (*Figure 2A*). However, also in line with our in vivo data, the two network patterns were not fully independent – plastic changes initiated in the network by means of persistent gamma activity altered the subsequent SWR pattern (*Figure 2A and B*). The p-SWRs exhibited a significantly increased area (by 69.7 ± 15.1%, n = 30, p<0.0001, *Figure 2C*) and a small decrease in incidence (to 0.90 ± 0.10 Hz, n = 30, p<0.0001). In addition, we found a slight but significant increase in average ripple number (by 8.5 ± 2.5%, n = 12, p=0.0065) and an elevated oscillatory ripple frequency (by 4.1 ± 1.8%, n = 12, p=0.041). These changes were accompanied by a significant increase of post-gamma sharp wave amplitude (by 44.6 ± 19.3%, n = 12, p=0.042) and a non-significantly altered sharp wave duration (increased by 2.5 ± 2.9%, n = 12, p=0.41). In good agreement with these data, gamma oscillations induced by bath application of carbachol (20 µM), an alternative drug to trigger persistent gamma oscillations based on a different mechanisms (*Fisahn et al., 1998*; *2002*; *Hajos et al., 2004*), also resulted in a significant increase in SWR area (by 21.0 ± 4.2%, n = 16, p=0.0002, see also *Zylla et al., 2013*), indicating that gamma activity itself, and not the pharmacological agent, is responsible for the network alterations. Moreover, we found a highly significant positive correlation

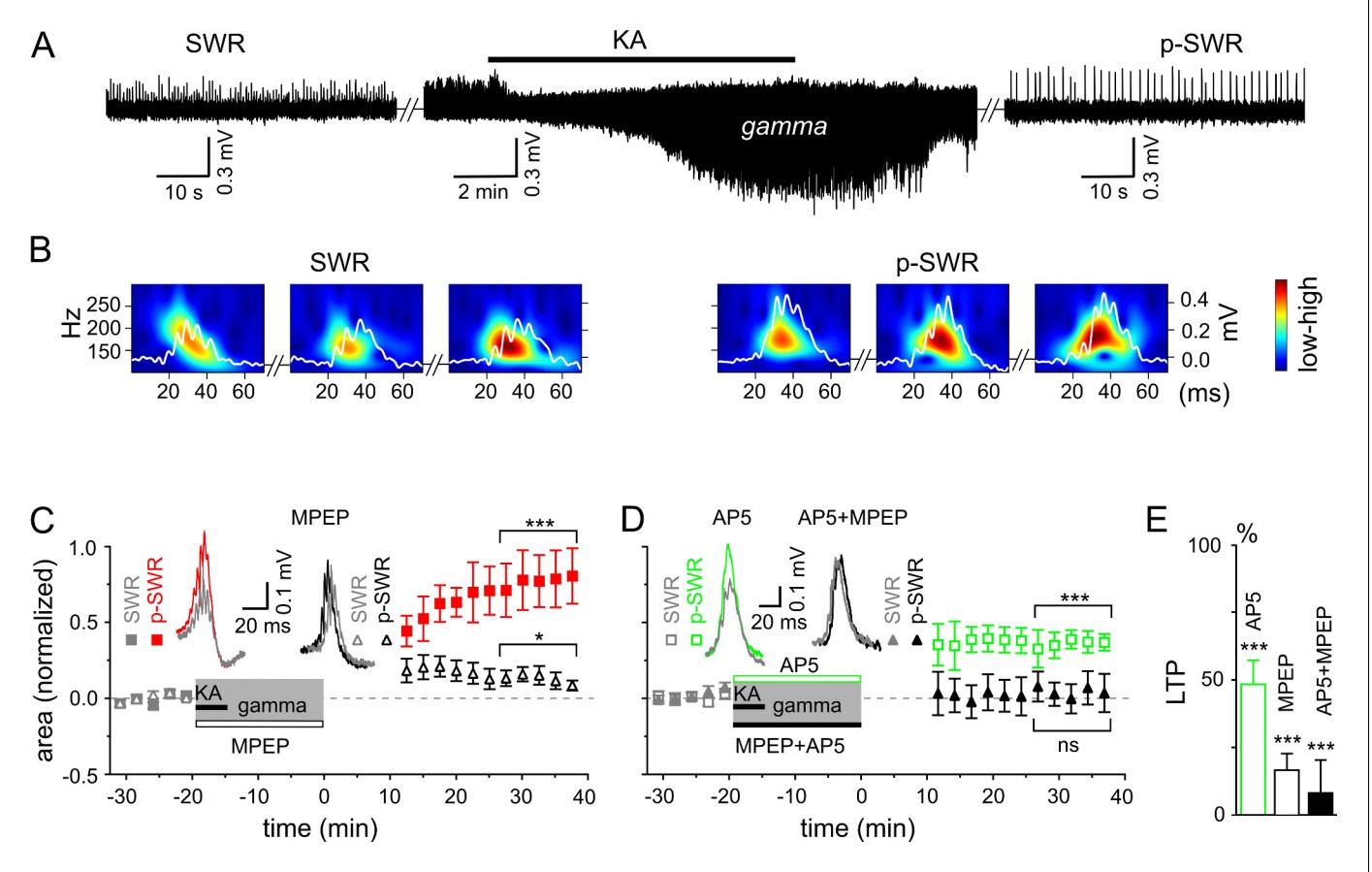

**Figure 2.** Gamma rhythms promote long-lasting alterations in the network activity. (**A**) SWRs recorded in the stratum pyramidale of the CA3 region occurred spontaneously (left), disappeared shortly after bath application of KA (middle) and reappeared with a significantly higher amplitude after KA washout (right). (**B**) Example of the wavelet transform (color-coded power spectral density) for three consecutive highlighted SWRs (white trace) before (SWR) and after (p-SWR) intermediate gamma oscillations. (**C**) p-SWR areas (red-filled squares) increased significantly compared to the SWR areas (gray-filled squares). MPEP (50 μM) administration largely prevented p-SWR area increase (black, open triangles). The time courses of drug applications are depicted schematically. The horizontal lines mark the p-SWR data points used for statistical analyses. The significance stars compare the pre-gamma data with the marked post-gamma data. The number of asterisks indicates the significance level (Student's t-test). Insert, examples of SWR (gray) and p-SWR (red and black) without (left) and with MPEP (right) administration during gamma rhythms. (**D**) Effects of AP5 (50 μM, green open squares) and MPEP+AP5 (black filled triangles) on SWR area increase. Insert, examples of corresponding SWR (gray) and p-SWR (green and black). (**E**) Gamma oscillation-induced SWR area increase (LTP) is reduced significantly by administration of AP5 (green open bar), MPEP (black open bar) and MPEP+AP5 (black filled bar).

of the network gamma oscillations (power×duration) with the SWR-area increase (R = 0.58, n = 30, p=0.0007) and no correlation with the SWR incidence (R = −0.28, n = 30, p=0.14). In line with this, in a few cases where KA application failed to introduce gamma frequency oscillations, no changes in SWR-area were observed (reduction by 6.0 ± 19.4%, n = 8, p=0.35). Notably, in comparison to KA, carbachol-triggered gamma oscillations exhibited less spectral power (p=0.007) and accordingly induced smaller SWR area changes (p=0.02), suggesting an activity-dependent mechanism of oscillation-induced neuronal network plasticity. Furthermore, we also found a reinforcing effect of gamma rhythms on subsequent gamma episodes (*Figure 3*). Together, these results emphasize the general potential of gamma oscillations to modify network activities.

To reveal the molecular mechanism underlying the network plasticity we examined the effects of activation of metabotropic glutamate (mGluR)5- and/or N-methyl-D-aspartate receptors (NMDAR) that have been proposed to play an important role in hippocampal synaptic plasticity and memory (*Zalutsky and Nicoll, 1990*; *Nakazawa et al., 2002*; *Naie and Manahan-Vaughan, 2004*). Administration of mGluR5 antagonist, 2-methyl-6-(phenylethynyl)pyridine (MPEP) largely reduced the SWR

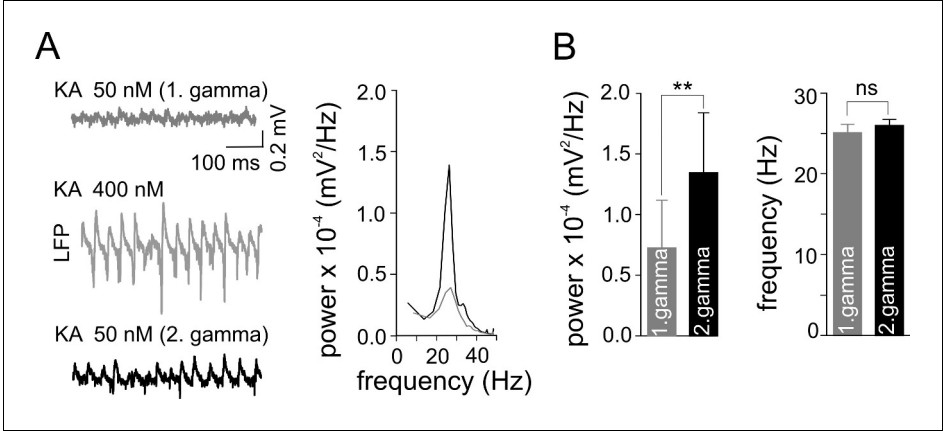

**Figure 3.** Gamma frequency oscillations promote changes in network activity. (**A**) Brief 'weak' field gamma episodes were induced by bath application of 50 nM KA. After this test period, 'conventional' gamma frequency oscillations were induced by 400 nM KA application, followed by KA washout achieving a complete cessation of oscillatory gamma activity. In a third step, the network behavior was again tested with low KA concentration as applied in the first step. Left, a 'weak' gamma episode (top, gray: 1st 'weak' gamma) becomes significantly stronger (bottom, black: 2nd 'weak' gamma) after 'conventional' ('KA 400 nM') gamma oscillations. Right, corresponding spectral analysis of 1st and 2nd 'weak' gamma. (**B**) Summary bar charts of peak power and frequency obtained before (1st 'weak' gamma) and after 'conventional' gamma (2nd 'weak' gamma). Spectral power of gamma rhythms increased from $0.72 \times 10^{-4} \pm 0.39 \times 10^{-4}$ mV$^2$/Hz to $1.34 \times 10^{-4} \pm 0.50 \times 10^{-4}$ mV$^2$/Hz (n = 10, p=0.003) while frequency remained unchanged, illustrating that the intervening gamma episode has a reinforcing effect.

area increase (by $83.4 \pm 6.2\%$, n = 16, p<0.0001, *Figure 2C and E*). A similar, albeit less pronounced effect was observed for the NMDAR antagonist, DL-2-Amino-5-phosphonopentanoic acid (AP5) (reduction by $51.7 \pm 8.9\%$, n = 13, p<0.0001, *Figure 2D and E*). Finally, this form of plasticity was abolished by joint application of the mGluR5 and NMDAR antagonists (reduction by $91.8 \pm 12.2\%$, n = 14, p<0.0001, *Figure 2D and E*). Taken together, these results suggest that in the hippocampal area CA3, gamma frequency oscillations influence the subsequent network activity through mGluR5- and NMDAR-dependent mechanisms.

## Gamma frequency oscillations support long-lasting synaptic plasticity in the CA3 network

We next studied activity-dependent alteration in synaptic transmission, defined here by synaptic strength, in CA3 PC by examining long-lasting changes in p-SWR-associated excitatory and inhibitory synaptic currents (p-EPSCs and p-IPSCs, *Figure 4*). Our data demonstrate that parallel to the field p-SWR (*Figure 2*), cells held in current-clamp mode during gamma oscillations (see Materials and Methods) exhibit long-lasting increase in the area of p-EPSCs (by $104.2 \pm 21.2\%$, n = 14, p=0.0003, *Figure 4B*). The change in the EPSC strength correlates positively with the magnitude of SWR-area increase (R = 0.56, n = 14, p=0.040). Conversely, p-EPSCs recorded in PCs held in voltage-clamp mode at $-70$ mV during gamma oscillations decreased rather than increased (by $40.6 \pm 7.7\%$, n = 13, p=0.0002, *Figure 4B*), suggesting that the increase of EPSC area depend on PC postsynaptic depolarization. The p-EPSC in voltage-clamp mode might also be affected by altered inhibition (see below).

To clarify the underlying molecular mechanism, we compared the properties of these currents in the presence and absence of mGluR5 antagonist MPEP. Similar to the effects on the LFP, bath application of MPEP strongly reduced the increase in p-EPSC area of CA3 PCs (reduction by $90.3 \pm 7.1\%$, n = 6, p=0.0001, *Figure 4B*). These data provide direct evidence for an mGluR5-dependent increase of the excitatory synaptic strength onto PCs as an effect of an intermediate gamma episode. Close temporal correlation existed between these changes and the network alterations described above.

We further investigated whether oscillatory gamma activity-dependent modification of the hippocampal network also includes alteration in inhibitory synaptic strength. In contrast to the strongly

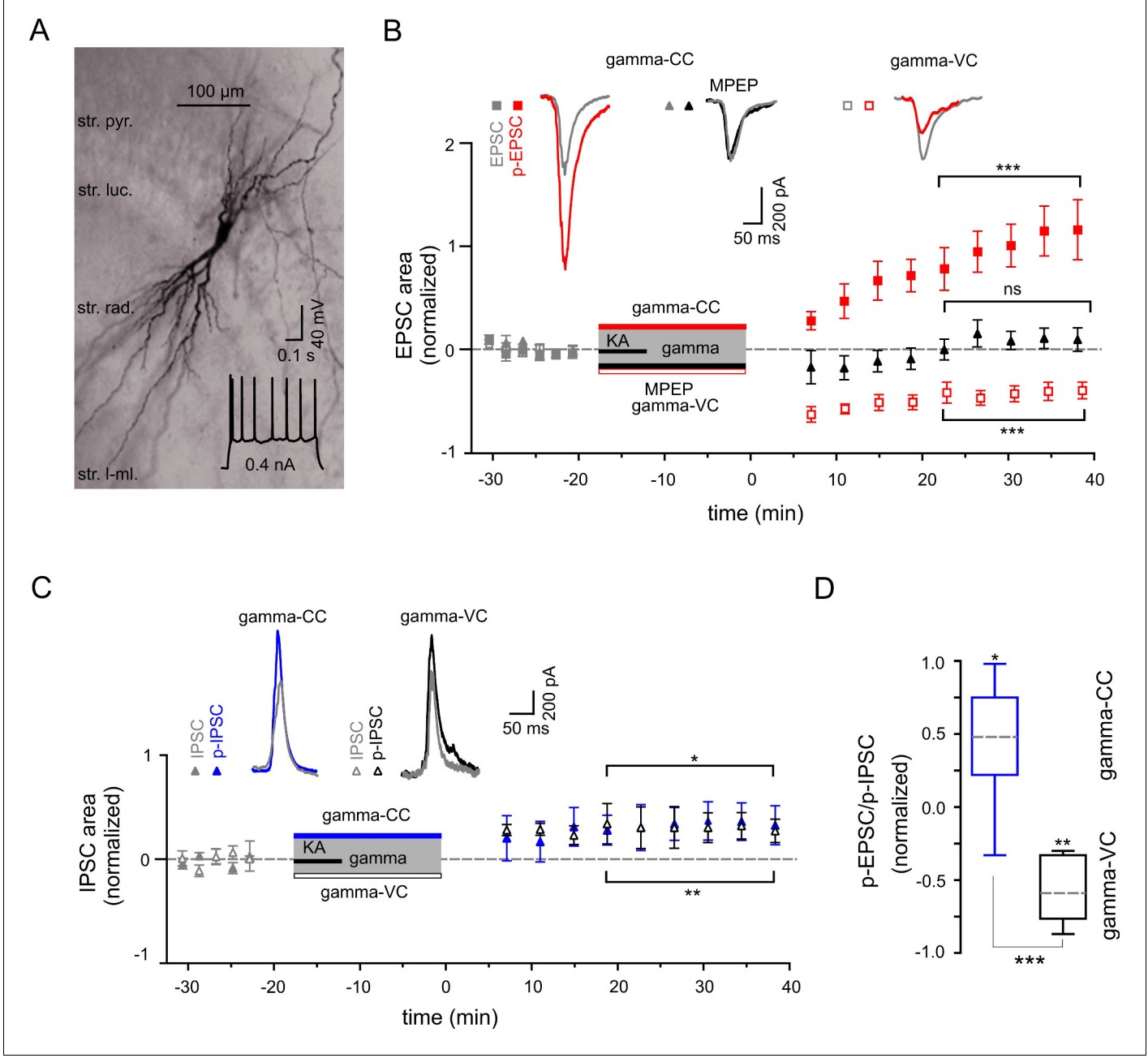

**Figure 4.** Gamma frequency oscillations support long-lasting synaptic plasticity. (**A**) Light micrograph of an example CA3 PC. Insert: The regular firing pattern of this PC in response to a depolarizing current injection. (**B**) The area of SWR-associated EPSC increases significantly after gamma frequency oscillations (gray and red filled squares for EPSCs and p-EPSC, respectively). Administration of MPEP (50 μM) prevents the increase of p-EPSC (gray and black filled triangles illustrate EPSC and p-EPSC, respectively). Holding PCs in voltage clamp configuration at −70 mV during gamma frequency oscillations leads to a significant decrease in EPSC area (gray and red open squares for EPSC and p-EPSC, respectively). The significance stars compare the pre-gamma data with the marked post-gamma data. Insert: Representative examples of EPSC (gray) and p-EPSC recorded without (left, red) and with MPEP (middle, black), as well as using voltage clamping of cells during gamma rhythms (right, red, gamma-VC). (**C**) SWR-associated IPSCs exhibit a moderate increase in area in PCs held in both current- (filled gray and blue triangles) and voltage-clamp mode (open gray and black triangles for IPSC and p-IPSC, respectively) during gamma rhythms. Inserts: Representative examples of corresponding IPSC (gray) and p-IPSC (gamma-CC, blue and gamma-VC, black). (**D**) Contra-directional change in p-EPSC to p-IPSC ratio for PCs held in current- (gamma-CC) *vs.* voltage-clamp mode (gamma-VC) during gamma rhythms (normalized to pre-gamma values).

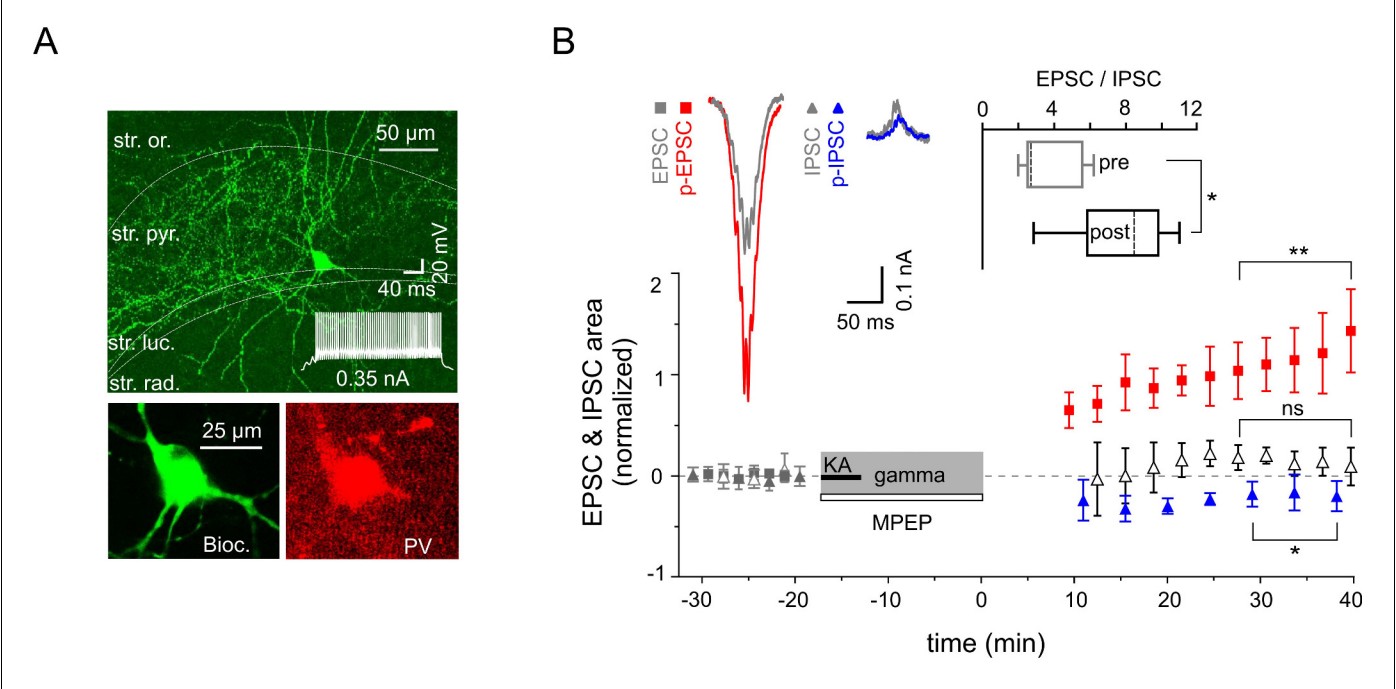

**Figure 5.** Increased excitability of fast spiking PV-expressing interneurons. (**A**) Top: A confocal image of a typical PV-expressing IN filled with biocytin. Insert: Fast firing pattern in response to depolarizing current injection. Bottom: Immuno-reactivity of the biocytin filled cell for PV. (**B**) Normalized EPSC (gray squares), IPSC (gray triangles), p-EPSC (red squares) and p-IPSC (blue triangles) recorded from PV-positive INs held in current-clamp mode during gamma rhythms. Administration of MPEP (50 μM) prevents the increase of p-EPSC (gray and black open triangles illustrate EPSC and p-EPSC, respectively). The significance stars compare the pre-gamma data with the marked post-gamma data. Inserts: left, representative examples of EPSC and IPSC (gray) with corresponding p-EPSC (red) and p-IPSC (blue); right, the EPSC-to-IPSC ratio before (pre) and after (post) gamma rhythms demonstrates that the excitability significantly increases in PV-positive INs.

potentiated p-EPSC, p-IPSCs showed a less pronounced but still significant increase (32.8 ± 12.9%, n = 8, p=0.039, *Figure 4C*). Altogether, p-EPSC and p-IPSC alterations resulted in a significant increase in the PC EPSC-to-IPSC ratio (by 44.4 ± 14.6%, p=0.019, *Figure 4D*). In stark contrast to the EPSC potentiation, changes in IPSCs were independent of postsynaptic activation. PCs held in voltage-clamp mode during gamma oscillations exhibited a similar IPSC increase of 30.3 ± 7.3% (n = 6, p=0.0089, *Figure 4C*), indicating activity-dependent changes in presynaptic inhibitory INs.

## Gamma rhythms promote cell type-specific synaptic plasticity in CA3 interneurons

To further elucidate the cellular mechanisms underlying the differential alterations in PC excitatory and inhibitory synaptic strength we examined the gamma frequency-dependent changes in two major inhibitory cell types: fast spiking PV-expressing INs targeting perisomatic or proximal dendritic domains of PCs and regular spiking CCK-containing perisomatic targeting cells. Fast-spiking PV-positive cells (*Figure 5A*) showed strong potentiation of EPSCs (increased by 89.8 ± 20.2%, n = 12, p=0.0010), whereas IPCSs decreased slightly but significantly (n = 5, p=0.038, *Figure 5B*). Combined, the latter translated into a significant rise in the EPSC-to-IPSC ratio (from 3.78 ± 0.83 to 7.71 ± 1.39, n = 5, p=0.012, *Figure 5B*). Similarly to the effect on PCs, application of mGluR5 antagonist MPEP strongly reduced the increase in p-EPSC area of PV-expressing INs (reduction by 84.6 ± 4.2%, n = 5, p=0.0001, *Figure 5B*). Markedly different, even inverse alterations were observed in the regular firing CCK-containing perisomatic targeting INs (*Figure 6A*): IPSCs were increased (by 61.01 ± 16.0%, n = 5, p=0.019), whereas EPSCs showed no change (reduction by 0.1 ± 10.6%, n = 8, p=0.99). As such, the EPSC-to-IPSC ratio was significantly reduced in these INs (from 2.14 ± 0.39 to 1.21 ± 0.26, n = 5, p=0.024, *Figure 6B*). These data, quantified by alterations in the EPSC-to-IPSC ratio, demonstrate that gamma frequency oscillations induce cell type-specific synaptic plasticity in

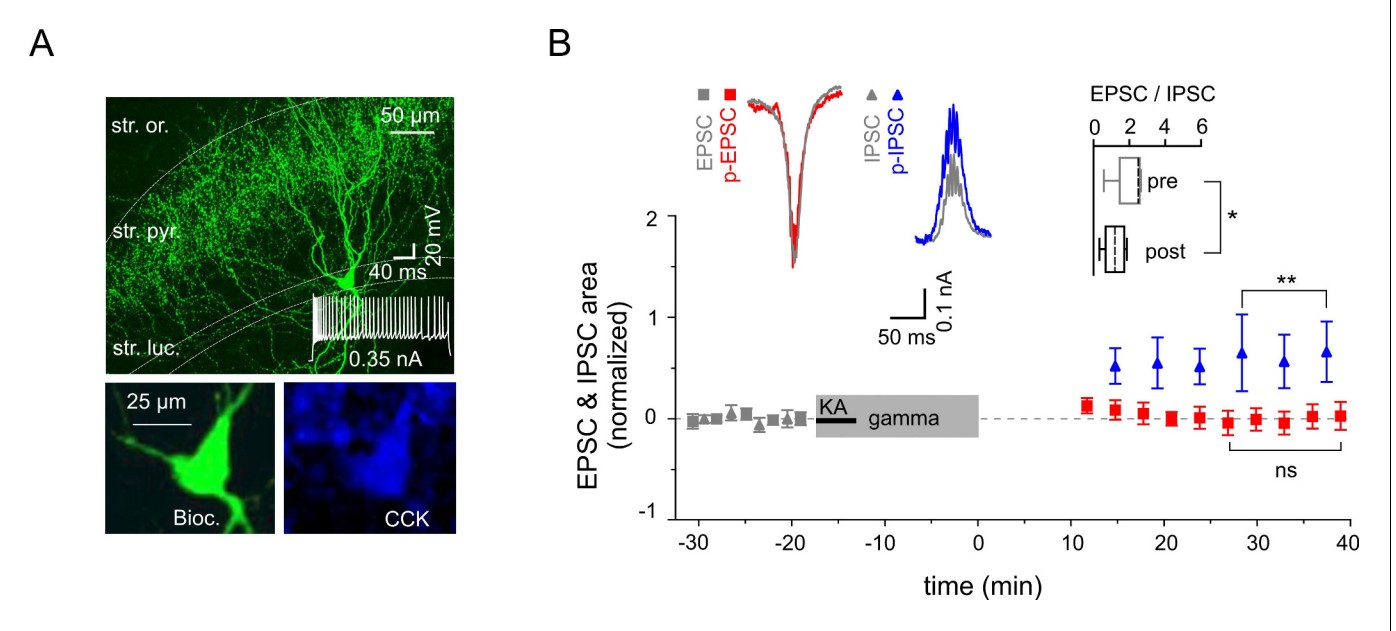

**Figure 6.** Reduced excitability of regular spiking CCK-expressing interneurons. (**A**) Top: A confocal image of a typical CCK-expressing IN filled with biocytin. Insert: Regular firing patterns in response to depolarizing current injection. Bottom: Immuno-reactivity of the biocytin filled cell for CCK. (**B**) Normalized EPSC (gray squares), IPSC (gray triangles), p-EPSC (red squares) and p-IPSC (blue triangles) recorded from CCK-expressing INs held in current-clamp mode during gamma frequency oscillations. The significance stars compare the pre-gamma data with the marked post-gamma data. Inserts: Left, representative examples of EPSC and IPSC (gray) with corresponding p-EPSC (red) and p-IPSC (blue); right, the EPSC-to-IPSC ratio before (pre) and after (post) gamma rhythms demonstrates that the excitability significantly decreases in CCK-positive INs.

the CA3 inhibitory network, with enhanced net excitation of PV-expressing INs, but reduced activation of CCK-expressing INs.

## Discussion

We have demonstrated that gamma frequency oscillations induce activity-dependent and cell type-specific synaptic plasticity in hippocampal area CA3. Moreover, our results illustrate the impact of oscillatory gamma activity on SWRs, a network state associated with the process of memory consolidation (*Girardeau et al., 2009*; *Jadhav et al., 2012*). The plastic changes require mGluR5 mediated activation, indicating that this receptor might be critically involved in memory processing.

We found that SWRs in vivo displayed a significantly enlarged area after a running episode, indicating a reinforcing effect of the running associated theta-nested gamma oscillations. Our data are well in line with a recent publication (*Bittner et al., 2015*) demonstrating that the induction of new place-fields initiated during active running results in altered neuronal activity during subsequent SWRs. While the oscillatory theta component does not seem to be essential for the induction of plastic changes in the hippocampus in vivo (*Brandon et al., 2014*), gamma rhythms are thought to constitute time windows of synchronized neural activity that promote spike-time-dependent synaptic plasticity (*Axmacher et al., 2006*) and enhance signal transmission (*Sohal et al., 2009*). In line with this, our in vitro results clearly demonstrate that a gamma frequency episode significantly affects subsequent network activities including gamma oscillations and SWRs. The gamma activity-induced effect (SWR area increase) was independent of the pharmacologic agent (KA *vs.* carbachol) used for their induction, but did correlate with the presence and power of gamma frequency oscillations. Indeed, activity dependent increase of SWR amplitude was shown by high-frequency electrical stimulation (*Behrens et al., 2005*). Thus, our results provide comprehensive data that the gamma oscillations and not the pharmacologic agents themselves (*Zylla et al., 2013*) are responsible for the observed network plasticity.

Different forms of plasticity have been described for the three excitatory input systems converging on CA3 PCs: mossy fibers and associational-commissural (A/C) and perforant path (PP) projections (*Urban and Barrionuevo, 1996*; *McMahon and Barrionuevo, 2002*; *Nakazawa et al., 2002*; *Kobayashi and Poo, 2004*; *Nicoll and Schmitz, 2005*; *Rebola et al., 2011*). However, crucially, the latter publications and similar studies on hippocampal neuronal plasticity have certain methodological limitations. First, they were usually elicited by high-frequency electrical stimulation of neurons providing afferent input, whereas the predominant firing rate of CA3 PCs and their afferent neurons in vivo is far less frequent (*Hahn et al., 2007*; *Jung and McNaughton, 1993*). Second, electrical stimulation was usually limited to one of these inputs, whereas, in the intact hippocampal network, individual inputs do not act in isolation, but converge onto postsynaptic cells. Thus, during physiological activity patterns, such as gamma frequency oscillations, different inputs to CA3 PCs may act synergistically, with their joint activity resulting in a specific alteration of synaptic strength. Consequently, the here investigated oscillatory pattern might constitute a more physiological paradigm that can elucidate network-dependent mechanisms of synaptic plasticity. With our approach, we reveal a unique role of gamma frequency oscillations in activity-dependent modification of hippocampal network. Our results highlight this oscillatory network rhythm as a fundamental mechanism to induce synaptic plasticity and a potential primary driving force for memory processing. Nevertheless, the specific role of an individual input for gamma-dependent plasticity in hippocampal network remains to be clarified.

Our data lend further support to the hypothesis that, overall, the two major memory relevant oscillatory patterns, gamma frequency oscillations and SWRs that are generated during different behavioral states in freely moving animals (*Chrobak et al., 2000*), can be considered two 'competing', mutually exclusive network states: spontaneous occurring SWRs disappeared shortly after onset of gamma rhythms and reappeared after their termination, both in vivo and in vitro. However, these two network patterns are not fully independent: plastic changes initiated in the network during persistent gamma activity were reflected in a subsequent altered SWR activity (*Figure 1* and *Figure 2*). Consistent with the here observed tight link of gamma oscillations and SWR activity, sleep-dependent memory consolidation is associated with increased gamma activity (*Ognjanovski et al., 2014*) and cells active during exploratory behavior exhibit enhanced SWR-associated EPSCs in subsequent slice preparations (*Mizunuma et al., 2014*).

Our data suggests that mGluR5 is a key component of the process underlying the observed plastic changes in the hippocampal CA3 network. In line with our findings, impairment of both LTP and spatial learning as well as place field encoding of novel environments induced by mGluR5 antagonists have been reported (*Naie and Manahan-Vaughan, 2004*; *Zhang and Manahan-Vaughan, 2014*). Group I mGluRs, comprising of mGluR5 and mGluR1, are preferentially expressed postsynaptically in CA3 PC dendrites (*Shigemoto et al., 1997*). Even though NMDAR might be involved, our results show that mGluR5 is more central in gamma network oscillation-induced synaptic plasticity. The effect can only partially be explained by NMDAR-modulation, with mGluR5 obviously exerting a more complex impact on the neuronal network dynamics, affecting both PC and IN activity. Interestingly, dysregulation of mGluR5 has been reported in several profound neurological disorders, such as schizophrenia (*Conn et al., 2009*; *Nickols and Conn, 2014*), autistic spectrum disorders (*Williams, 2012*) and fragile X syndrome (*Michalon et al., 2012*), altogether pointing towards a pivotal regulatory function for this receptor. Our results highlight mGluR5 now in the general context of memory processing and neuronal plasticity.

In contrast to the postsynaptically mediated potentiation of PCs excitatory currents, inhibitory currents only underwent minimal changes that were independent of postsynaptic activation. These differences could be explained by a cell-specific, directionally biased synaptic plasticity at PC-IN and IN-IN synapses as demonstrated for two major types of GABAergic inhibitory cells, PV- and CCK-expressing INs. Gamma network oscillations alter synaptic strength within PV-expressing INs in favor of excitation (*Alle et al., 2001*), while CCK-expressing INs are subject to stronger inhibition, as demonstrated by the EPSC-to-IPSC ratio analysis. Importantly, inhibition provided by these two types of GABAergic cells is not uniform. Fast-spiking PV-expressing INs mediate a rapid, phasic-form of inhibition, which contributes to the precise timing of neuronal synchronization and emergence of network oscillations (*Gloveli et al., 2005a*; *Sohal et al., 2009*; *Schlingloff et al., 2014*). In contrast, regular firing CCK-expressing INs mediate slower inhibition (*Hefft and Jonas, 2005*; *Daw et al., 2009*) and modulate excitability in cortical networks in a behavioral state-dependent manner. Thus,

the two IN types mediate distinct forms of inhibition and could contribute differentially to cortical network activity. Our data now suggest that divergent forms of synaptic plasticity observed in these two IN types could result in a reduced tonic but increased phasic inhibition onto PC. These changes in turn might lead to enhanced network excitability and promote synaptic plasticity within the cortical circuits.

In summary, we conclude that gamma frequency oscillations represent a network state responsible for activity-dependent and cell type-specific synaptic plasticity, interlinking two memory-relevant network patterns, namely, gamma rhythms and SWRs.

## Materials and methods

### Animals

Experiments were performed on P27-P33 (in vivo) and P18-P23 (in vitro) C57/Bl6 mice. All animal procedures were approved by the Regional Berlin Animal Ethics Committee (Permits: G0151/12 and T 0124/05) and were in full compliance with national regulations.

### Local field potentials in vivo

We recorded LFP from head-fixed mice, a well-established approach that allowed us to conduct prolonged running episodes. Mice were first implanted with a head-holder and a recording chamber (1.5% isoflurane anesthesia) and then habituated to a spherical treadmill for around 12 days. Afterwards, a small craniotomy (approx. 2.3 mm rostro-caudal and 2.5 mm lateral from bregma; 1.5% isoflurane anesthesia) was performed inside the recording chamber and the exposed area covered with a layer of silicone elastomer (Kwik-Sil, World Precision Instruments). The mouse was allowed to recover for at least 2 hr before the recording session started (one recording session per mouse).

LFP from the left hippocampus were recorded with glass pipettes, while we were using the control of behavioral expression without a task-specific reward to target certain network patterns. To investigate the impact of a theta-nested gamma episode on SWRs we first waited for a prolonged resting period with a quietly sitting mouse on the spherical treadmill allowing us to record spontaneous SWR activity. Then, once the mouse had begun to move independently, a pressurized air stream was applied to the bottom of the Styrofoam ball, resulting in a smooth ball rotation that encouraged a running behavior accompanied by theta-nested gamma oscillations. If the mouse stopped running and began to balance the air-supported ball instead, we accelerated the ball slightly until running behavior was restored, maintaining an activity phase around 3 min in total depending on the actual running performance. Turning off the air pressure usually terminated the running behavior and initiated another resting period, once again accompanied by spontaneous occurring SWRs. In order to reduce the stress level while maintaining the attentional component, some dummy runs were performed prior to the final recording session.

### Matlab analysis of in vivo data

All in vivo LFP data were analyzed in Matlab (MathWorks Inc., Natick, Massachusetts) by means of custom-made routines. We compared the SWR areas of two time periods (120 s each) ending 10 s before and starting 30 s after the prolonged running episode. LFP recordings were divided into a sharp wave (filtered 2–50 Hz) and a ripple (filtered 100-300 Hz) trace. We used the ripple trace to automatically preselect SWRs based on a voltage and a spectral threshold criterion. In detail, we first used a voltage threshold (mean plus six standard deviations of event-free recording) for a primary selection of individual ripples and grouped adjacent single ripples to a ripple event. We then took 70 ms cutouts of the event-free recording preceding those ripple events and calculated each maximum absolute wavelet coefficients of the complex Morlet wavelet transform (27 wavelet scales, 20 kHz sampling rate). We used the mean plus one standard deviation of this distribution as a spectral threshold criterion and discarded all ripple events, in which the maximum absolute wavelet coefficient did not exceed the threshold value. The preceding and subsequent local minima in the sharp wave trace were used to automatically identify SWR start and end points. However, the bandpass (2–50 Hz) filtered in vivo sharp wave trace still exhibited a remarkable variation, leading to incorrect boundaries in some cases. Consequently, the sharp wave, ripple and original recording traces were scrutinized by eye (*Forro et al., 2015*). We rejected erroneously detected SWRs and manually

adjusted the automatically identified start and end points if required. Finally, the SWR area was defined in the sharp wave trace as the area beneath the curve enclosed by those start and end points. However, comparing the uncorrected automatically identified SWRs we also obtained a statistically significant difference. Spectral power densitiy of gamma frequency oscillations were determined with an Welch algorithms (pwelch) and the complex Morlet wavelet transform (cmor2-1) was used to display SWR (bandpass filter 100–300 Hz, 134 wavelet scales, 20 kHz sampling rate).

## Slice preparation

The animals were anesthetized with inhaled isoflurane, decapitated and the brains removed. Tissue blocks containing the hippocampal formation were mounted on a Vibratome (Leica VT1200) in a chamber filled with ice-cold artificial cerebrospinal fluid (ACSF). Transverse hippocampal slices were cut at 400 µm thickness and incubated for at least 1 hr in a holding 'interface' chamber (continuously oxygenized with carbogen and perfused with ACSF at ~2 mL/min) and then transferred to the recording 'submerged' chamber (perfused at a rate of 6 mL/min), both at 33 ± 1°C. The solution used during cutting, incubation and recording contained (in mM): NaCl, 129; KCl, 3; $NaH_2PO_4$, 1.25; $CaCl_2$, 1.6; $MgSO_4$, 1.8; $NaHCO_3$, 21; glucose, 10; saturated with 95% $O_2$ and 5% $CO_2$, pH 7.4; 290–310 mOsm.

## Local field potentials in vitro

LFP were obtained from the stratum pyramidale of the hippocampal CA3 area. KA (400 nM, unless indicated otherwise) or carbachol (20 µM) were applied in the bath to induce network gamma frequency oscillations. The SWR oscillations occurred spontaneously, disappeared shortly after bath application of KA or carbachol and reappeared within a few minutes after their washout. mGluR5 and/or NMDAR activation was blocked by MPEP (50 µM, Tocris Bioscience) and/or AP5 (50 µM, Tocris Bioscience). MPEP and/or AP5 were launched simultaneously to KA, but continued throughout the entire oscillatory gamma network episode.

Field oscillations were low pass filtered at 5 kHz, digitized at 10 kHz (Digidata 1440A, Axon Instruments) and analyzed with the pClamp software package (notch filter 50 Hz; Axon Instruments). Oscillatory peak power and frequency was determined by averaging several consecutive fast Fourier transforms (FFT). SWRs were identified and the area under curve calculated (pClamp software, Axon Instruments). A Student's t-test was used for statistical comparisons unless stated otherwise; differences were considered significant if $p < 0.05$. Average values are expressed as mean ± SEM. Spearman's rho was used to assess statistical dependence. Pre- and post-gamma data values were normalized to the mean of all prior gamma data. The EPSC-to-IPSC ratio was used to assess the net changes in cellular excitability. We further analyzed the spectral components of the LFPs with custom routines written in Matlab. Signals were zero-phase digital filtered from 2–300 Hz using a Butterworth filter, 50 Hz components including their harmonics were removed using a second-order infinite impulse response notch filter. A complex Morlet wavelet transform (cmor2-1) was used to display SWRs (bandpass filter 100–300 Hz, 134 wavelet scales, 20 kHz sampling rate).

## Whole-cell recording in vitro

The patch-clamp recordings were obtained from PCs and INs of hippocampal CA3 area visualized by infrared differential interference contrast video microscopy. The intrinsic and firing properties of cells were measured in whole-cell current-clamp mode as described previously (*Gloveli et al., 2005a*). In order to follow the Hebbian plasticity rules, during gamma frequency oscillations, cells were recorded in current-clamp mode enabling them to generate action potentials. In an additional set of experiments, the PCs were held in voltage-clamp mode at −70 mV during gamma activity to prevent their depolarization. Whole-cell recording pipettes (3–5 MΩ) were filled with a solution containing (in mM): K-gluconate, 135; KCl, 5; ATP-Mg, 2; GTP-Na, 0.3; HEPES, 10; plus biocytin, 0.5% (pH 7.4 and 290 mOsm). A Multiclamp 700B amplifier and pClamp software (Axon Instruments) were used for current- and voltage-clamp recordings. The holding potential in voltage-clamp mode was either −70 mV or 0 mV to record the EPSCs and IPSCs, respectively. The areas under curve were calculated for EPSCs and IPSCs and the EPSC-to-IPSC ratios were determined. The seal resistance before establishing whole-cell mode was ≥2 GΩ. The series resistance (range 12–18 MΩ) was not compensated, but was repeatedly monitored during the experiment by measuring the amplitude of

the capacitive current in response to a −10 mV pulse. Experiments, in which the series resistance increased by >20% were discarded. Signals were low-pass filtered at 5 kHz, digitized at 10 kHz (Digidata 1440A) and analyzed using pClamp software.

The firing properties of IN [fast (>100 Hz), non-accommodating *vs.* regular] were studied using intrasomatic current injection (0.5 nA). Electrophysiological identification was confirmed by post hoc immunostaining and biocytin staining.

### Immunolabeling

For immunolabeling of interneurons, slices were immersed overnight in a fixative solution containing 4% paraformaldehyde (PFA) in 0.1 M phosphate buffer (PB), washed three times in 0.1 M PB and subsequently in 0.025 phosphate-buffered saline (PBS; pH 7.3). Slices were then incubated in PBS containing 1% Triton X-100, 10% goat serum and Mouse on Mouse (M.O.M) blocking reagent (2 drops per 2.5 ml solution) for 1 hr at room temperature (RT). To visualize PV- and CCK-containing cells, we used antibodies against PV (mouse, Swant, Marly, CH) and CCK (mouseCURE, Los Angeles, CA) diluted 1:5000 in PBS containing 5% goat serum and 1% Triton X-100. Slices were incubated with primary antibodies for 48 hr at RT. After rinsing three times in PBS, sections were incubated in the PBS solution containing 0.5% Triton X-100, 5% goat serum, goat anti-mouse conjugated with (for PV) Alexa fluor 546 (Invitrogen Corporation, Carlsbad, CA) or (for CCK) Alexa fluor 568 (Invitrogen Corporation, Carlsbad, CA) diluted 1:500 or (for biocytin-filled neurons) Alexa fluor 647 (in some experiments 350) conjugated avidin diluted 1:500 (Invitrogen Corporation, Carlsbad, CA). Slices were mounted on glass slides in the glycerol-based, aqueous mountant Vectashield (Vector Laboratories) under coverslips at 48 hr after incubation with the secondary antibodies. Labeled cells were visualized using 20x and/or 60x objectives on a confocal microscope system (Leica). To examine the full extent of somato-dendritic compartments and axonal arborization, the intensity of Z-stack projections was optimized and the images were overlaid.

### Biocytin staining

Slices were processed as described previously in principle (*Dugladze et al., 2012*). For biocytin staining, slices with biocytin-filled cells were removed from the chamber and immersed overnight in a fixative solution containing 4% paraformaldehyde (PFA) in 0.1 M phosphate buffer (PB). Slices were washed three times in 0.1 M PB. The avidin–biocytin complex reaction (Vectastain ABC kit, Camon laboratory service) took place overnight at 4°C in the presence of 0.3% Triton X-100 (Sigma-Aldrich). Afterwards the sections were rinsed several times before development with 0.02% diaminobenzidine in 0.1 M PB. The reaction product was intensified with 0.5% OsO4 and sections were mounted and coverslipped. Stained cells were reconstructed with the aid of a Neurolucida 3D system (MicroBrightField, Inc).

### Additional Information

Matlab source code files for the calculation of FFT, Welch's spectrogram and the wavelet transformation are available on our homepage (https://glovelilab.wordpress.com).

## Acknowledgements

We thank Imre Vida for critically reading the manuscript and assisting with the immunostaining.

## Additional information

### Funding

| Funder | Grant reference number | Author |
|---|---|---|
| Deutsche Forschungsgemeinschaft | GL 254/5-2 | Tengis Gloveli |
| Bundesministerium für Bildung und Forschung | BCCN II, A3 | Tengis Gloveli |
| Einstein Stiftung Berlin | A-2013-176 | Tengis Gloveli |

| Deutsche Forschungsgemeinschaft | NeuroCure, Cluster of Excellence 257 | Tamar Dugladze |
| Deutsche Forschungsgemeinschaft | Research Unit FOR 2143 | Jörg RP Geiger |

The funders had no role in study design, data collection and interpretation, or the decision to submit the work for publication.

### Author contributions

SZ, PB, Acquisition of data, Analysis and interpretation of data, Drafting or revising the article; JS-T, CB, Acquisition of data, Drafting or revising the article; DS, Analysis and interpretation of data, Drafting or revising the article, Contributed unpublished essential data or reagents; JRPG, Drafting or revising the article, Contributed unpublished essential data or reagents; TD, TG, Conception and design, Acquisition of data, Analysis and interpretation of data, Drafting or revising the article, Contributed unpublished essential data or reagents

### Author ORCIDs

Tengis Gloveli, http://orcid.org/0000-0002-2209-375X

### Ethics

Animal experimentation: All animal procedures were approved by the Regional Berlin Animal Ethics Committee (Permits: G0151/12 and T 0124/05) and were in full compliance with national regulations. All surgery was performed under isoflurane anesthesia, and every effort was made to minimize suffering.

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
