## [Decision Letter]

Thank you for submitting your article "Cell-Specific synaptic Plasticity Induced by Network Oscillations" for consideration by *eLife*. Your article has been favorably evaluated by Eve Marder (Senior editor) and three reviewers, one of whom, Marlene Bartos, is a member of our Board of Reviewing Editors.

The reviewers have discussed the reviews with one another and the Reviewing Editor has drafted this decision to help you prepare a revised submission.

Summary:

Your study provides experimental evidence for the potentiation of excitatory synaptic inputs onto PV-interneurons in the hippocampal area CA3 during pharmacologically induced gamma oscillations in acute slice preparations. It further shows that network oscillations increase the strength of excitatory synaptic inputs which in turn lead to changes in sharp wave ripple complexes (SWRs). The group I metabotropic glutamate receptor 5 (mGluR5) and NMDA receptors are involved in this form of synaptic plasticity. Interestingly, excitatory inputs onto PV-expressing interneurons are enhanced but not the ones onto CCK cells suggesting an interneuron type-specific effect.

Essential revisions:

Two reviewers and the reviewing editor discussed your study on cell-specific synaptic plasticity induced by network oscillations. All reviewers viewed the study as 'important' because gamma oscillations are observed during the learning process. However, some main criticisms have been formulated which are listed below.

1) It remained still unclear how gamma oscillations affect the amplitude of SWRs. This criticism should be addressed by additional data analysis. One of the reviewers proposed the analysis of the correlation between the amplitude of gamma oscillations and the increase from pre-SWR to post-SWR. Moreover, the relationship between gamma and theta oscillations to frequency of SWR, the number of ripples per sharp wave and the duration of sharp waves should be analyzed.

2) The reviewers wished in some cases a higher number of experiments. In some cases only 4 experiments have been performed. Thus, increase the n to > 4.

3) The reviewers wished to know whether the magnitude of EPSC increases correlated with the magnitude of SWR increases.

4) The authors have presented data regarding the sensitivity of LFP and PC's to mGluR antagonists. A similar set of experiments are required for PV-interneurons. Addressing these points in a time window of maximal 2 months would be important to improve the study.

---

## [Author Response]

Essential revisions:

Two reviewers and the reviewing editor discussed your study on cell-specific synaptic plasticity induced by network oscillations. All reviewers viewed the study as 'important' because gamma oscillations are observed during the learning process. However, some main criticisms have been formulated which are listed below.

1) It remained still unclear how gamma oscillations affect the amplitude of SWRs. This criticism should be addressed by additional data analysis. One of the reviewers proposed the analysis of the correlation between the amplitude of gamma oscillations and the increase from pre-SWR to post-SWR. Moreover, the relationship between gamma and theta oscillations to frequency of SWR, the number of ripples per sharp wave and the duration of sharp waves should be analyzed.

We thank the reviewers for the suggestion of performing additional data analysis, which has allowed us to present an even deeper understanding of the underlying mechanisms. We first analyzed the impact of gamma oscillations on the SWR area (Figure 2) and frequency as suggested. We used the oscillatory power and the duration of network gamma oscillations (power×duration) to characterize the correlation of the SWR-area increase with the presence and power of gamma episodes. Indeed, we found a highly significant correlation of this parameter with the SWR-area change, well in line with our hypothesis of an activity-dependent mechanism. There was a negative, albeit not significant correlation coefficient, to the incidence of SWRs. We adjusted the main text accordingly (see Results section).

The fact that the internal state might not be constant for an extended time in awake behaving mice hampered the interpretation of the in vivo result (see Results section), including the analysis of the direct impact of gamma and theta oscillations on SWR properties. Therefore, we focused on the in vitro results and analyzed the relationship between gamma oscillations and the number of ripples and the amplitude and duration of sharp waves, as suggested. We found a significant increase of post-gamma sharp wave amplitude together with a virtually unchanged duration. Furthermore, there was a slight increase in the average number of ripples per sharp wave and an elevated oscillatory ripple frequency (see Results section).

2) The reviewers wished in some cases a higher number of experiments. In some cases only 4 experiments have been performed. Thus, increase the n to > 4.

We followed the reviewers’ suggestion and performed new experiments. All sample sizes of the revised manuscript are n>=5 as requested. Importantly, the additional experiments did not alter the conclusions and the statistics of our study, with the exception of the PV-positive IN p-IPSC values, which were significantly decreased, further accentuating the corresponding EPSC/IPSC-ratio increase. We changed the main text and the original Figure 3 and Figure 4 (new Figure 4 and Figure 5) accordingly.

3) The reviewers wished to know whether the magnitude of EPSC increases correlated with the magnitude of SWR increases.

We thank the reviewer for this suggestion. Indeed, we found a correlation between the magnitude of PC EPSC increases and the magnitude of SWR-area changes. These results are now included in the revised Results section.

4) The authors have presented data regarding the sensitivity of LFP and PC's to mGluR antagonists. A similar set of experiments are required for PV-interneurons. Addressing these points in a time window of maximal 2 months would be important to improve the study.

Following the reviewers’ request, we present new data on the effect of the mGluR5-antagonist MPEP on the EPSCs of PV-positive INs in the revised manuscript (see Results section and Figure 5). Similarly, to the effects on the LFP and PC EPSCs, bath application of this drug strongly reduced the increase in post-gamma EPSC areas of PV-positive INs. These data provide additional, direct evidence for an mGluR5–dependent effect on the neuronal network dynamic.